# Students' figurative communication of malaria messages, belief, norms, and practices in Oromia, Ethiopia: A qualitative content analysis approach

Kasahun Girma Tareke[1]*, Abdu Hayder[2], Firanbon Teshome[1], Zewdie Birhanu[1], Yohannes Kebede[1]

1 Department of Health, Behavior and Society, Institute of Health, Jimma University, Jimma, Ethiopia,
2 Department of Public Health, Mizan-Tepi University, Mizan-Tepi, Ethiopia

* kasahungirmadera@gmail.com

**Data Availability Statement:** The minimal data set is within the paper.

## Abstract

### Background

School engagement is an emerging strategy and proven potent vehicles for social and behavioral change communication (SBCC) intervention to prevent and control malaria. Little was known about the figurative speeches used in the malaria messages disseminated and communicated by school students. Therefore, this study evaluated the figurative speeches used in the poems to convey messages related to malarial perceptions, beliefs, norms and practices to prevent and control malaria.

### Methods

A qualitative content analysis was conducted to explore the figurative speeches used in malaria messages conveyed in poems produced by primary school students. Twenty poems were purposively selected from twenty schools across rural villages in five districts of Jimma Zone. Data were analyzed using ATLAS.ti version 7.1.4 software. The figurative speeches were presented using central themes and categories supported with quotations.

### Results

The predominantly used figurative speeches were simile, metaphor, personification and hyperbole. Simile was used to express the nature of anopheles mosquito, and sign and symptoms of malaria. The metaphor was used to express malaria, severity/seriousness of malaria and Insecticide-Treated Net (ITN); and also to express the relationship between persons ITN malpractice and its effect on their health. Personification was used to express the nature of anopheles mosquito and malaria. Finally, hyperbole was used to express nature of anopheles mosquito, severity of malaria and exaggerated effect of ITN and Indoor Residual Spraying (IRS).

**Funding:** The funders had no role in study design, data collection and analysis, decision to publish, or preparation of the manuscript.

**Competing interests:** The authors have declared that no competing interests exist.

**Abbreviations:** IRS, Indoor Residual Spraying; ITN, Insecticide-Treated Net; SBCC, Social and Behavioral Change Communication; WHO, World Health Organization.

## Conclusions

The students conveyed messages related to malarial perceptions, beliefs, norms and practices of the local community to prevent and control malaria through different types of figurative speeches. Therefore, conceptualizing the local norms, beliefs, values, perception and practices, and expressing in different figurative speeches to convey messages and convince the local community might be important to bring the desired or intended behavioral change.

## Introduction

Different strategic initiatives or approaches have been endorsed and recommended by the World Health Organization (WHO) and different implementing partners to reduce the public health impact of malaria [1–7]. A school-based social and behavioral change communication (SBCC) is one of the strategies, and proven potent vehicles to effectively and efficiently disseminate malaria messages to the community. The assumption is that the school community (students and teachers) act as a key agent to improve the capacity, knowledge, and decision-making skills to the community to promote health and prevent diseases, and also play a role in encouraging community-wide malaria prevention and control [1, 2].

School-based SBCC malaria activities include provision of training on malaria, conducting of peer learning/education, producing school mini-media materials, conducting peer discussions, provision of training focal teachers about in-school malaria prevention and control activities aiming to address behavioral factors related to knowledge, misconception, self -efficacy, perceived risk/severity, and practice, and ultimately reaching out to their families and neighbors with messages [2, 3, 8]. This type of intervention involves development of malaria messages in different forms, and dissemination and communication through mini-media, student clubs, peer education, group discussion, parent day, community meeting, and during school closing by school students, and other events organized by education and health offices [1, 2].

The messages were developed in the form of poem to disseminated information related human experiences, knowledge, self-efficacy, perceptions, perceived threat, thoughts, beliefs, views, imaginations, emotions, and practices related to malaria prevention and control strategies to the target audience in a more artistic way [1, 2, 8–10]. Moreover, it is natural and common to find various forms of figurative speeches in poems. It is used to express a certain word or phrase differently from the literal interpretation of language or the straightforward use of words for the sake of comparison, emphasis, clarity, freshness, special effect, colorful and forceful writing [11–13].

Simile and metaphor types of figurative speeches are used to make comparison between certain phenomena. Hyperbole is used to provide a dramatic effect expressing a phenomenon in an exaggeration/overstatement manner. A metonymy type of figurative speech is also sued to replace the name of something with closely related phenomenon. A paradox is used to convey a situation or statement that seems self-contradictory and even absurd, but may contain an insight into life. Personification is used to express assigning of human characteristic to non-humans. A figurative speech is also used to express a statement or situational meaning contradicted by the appearance or presentation of the idea or express magnitude of a statement by denying its opposite/understatement [1–13].

In the study setting, a school-based SBCC intervention was conducted since 2017 to prevent and control malaria, and the primary students were developed and disseminated malaria

messages in the form of poems to the community. Moreover, a study conducted in this setting indicated that the project contributed a significant change in comprehensive knowledge, message acceptance, practices of Insecticide-Treated Net (ITN) utilization, giving priority to children<5 years old and pregnant women, environmental cleaning specifically breeding site of mosquito, Indoor Residual Spraying (IRS) handling and early treatment-seeking for fever [14].

However, the students' figurative speeches expressed in the poems were not analyzed. Furthermore, to the knowledge of investigators, no published evidence was available that explored figurative speeches expressed in malaria message poems disseminated and communicated by primary school students to change the behavior of the community towards malaria prevention and control. Therefore, this study was explored and analyzed the figurative speeches expressed malarial message poems that are locally produced by the primary school students.

## Methods

### Study setting and period

The study was conducted in Jimma Zone, Oromia National Regional State, Ethiopia, from April to May, 2020. Jimma Zone has 21 districts, and 42 urban and 513 rural kebeles. The total population of Jimma Zone is estimated to be 3.2 million with the majority of the population living in rural area [15]. The data were collected from five districts: Limmu-Kosa, Botor-Tolay, Gera, Shebe-Sombo, and Nono-Benja. The districts were selected from the districts where school-based SBCC intervention was implemented. There were 15 districts with a total of 75 primary schools included in the intervention. This is because they are categorized under the high-medium-malaria-burden strata.

### Study approach

A qualitative content analysis was conducted to analyze the different types of figurative speeches used to express local perceptions, beliefs, emotions, knowledge, practices and values related to malaria prevention and control. Qualitative content analysis is used to analyze text data, and messages of communication materials to provide knowledge and understanding of the phenomenon under the study [12, 13, 16].

### Parent population, sample size and sampling technique

The intervention was conducted over the periods of 2017–2019 at five districts (Limmu-Kosa, Botor-Tolay, Gera, Shebe-Sombo, and Nono-Benja) which comprised a total of 75 primary schools. A lot of poems were produced, disseminated and communicated by students among these schools. However, only five schools were purposively selected for this study based on criteria's such as having high number of students enrolled in the school (ranged from 440–1450), being located at high malaria endemic areas, feasibility in terms of distance and active engagement and better involvement of students in producing poems [17]. A purposive sampling technique was used to select poems from school-based malaria SBCC documentation based on like richness, relevancy and data diversity pertinent to the research question. Specific criteria's used to select the poems were presence of at least three behavioral constructs (from knowledge, attitude, risk- perception, self-efficacy, response -efficacy, and practice) through roughly reading; presence and readability of the poems, length of the poems (i.e., at least two pages) and presence at least one type of figurative speech. Accordingly, 20 poems were selected from 20 selected schools across the districts. A sample of poems was selected from different schools across the districts and students grades to maximize data triangulation. The poems were collected equally from grade 5–8 [Table 1].

**Table 1. Sampling distribution poems produced at primary schools, target districts, Jimma zone, Oromia, Ethiopa, 2020.**

| Districts | Sample allocation | School-1 | School-2 | School-3 | School-4 |
|---|---|---|---|---|---|
| Shebe-Sembo | 4 | M/sedecha = 1 | Y/dogena = 1 | Mirgano = 1 | Kishe = 1 |
| Limmu-Kosa | 4 | Ambuye = 1 | Gumar = 1 | D/Gebana = 1 | C/Ifeta = 1 |
| Gera | 4 | K/Kindibit = 1 | Sedi = 1 | G/Challa = 1 | Dusta = 1 |
| Nono Benja | 4 | Illu = 1 | Ebicha = 11 | Amido = 1 | Kolatie = 1 |
| Botor Tolay | 4 | B/Adare = 1 | L/Botor = 1 | B/Barite = 1 | K/Boso = 1 |
| Total | 20 | | | | |

Note: Five poems were purposively selected from grade 5, 6, 7 and 8

## Data collection procedures

The poems were collected from 20 primary schools. Four individuals from graduate students (MPH) and BSc nurse were employed to collect the data. Half day orientation was given to them on the purpose of the study and data collection procedure. Data collectors travelled to the selected schools, and collected based on the selection criteria's. Then, according to the sampling distribution, the investigators had undergone further reading to pick more informative ones.

## Data analysis

A poem analysis approach was followed to analyze the data. First, reading of the poems was carried out to understand the impression of the poem. Repeat reading of the poems was also done to explore the figurative languages (simile, metaphor, personification, hyperbole) and proverbs. Then, two investigators (KGT and AH) developed a codebook manual after independently conducting a line by line coding of the poems and generating the codes, categories, sub-categories, and themes from the data (poems). Available profile (age, sex, grade, academic status, etc.) of the students was connected to poems. ATLAS ti 7.1.4 software was used to assist in organizing, managing, coding, editing, note taking and node/category manipulation of qualitative data in a more efficient manner. To enhance inter-coder reliability, the two coders reviewed, discussed and solved for differences in coding. Then, the poems were coded using the codebook manual to ensure code consistency and credibility was done. Next, potential sub-categories were developed by clustering codes. Categories and themes were developed by clustering sub-categories and categories, respectively which answer the research questions. Literature experts were consulted during analysis in order to classify types of figurative speeches emphasized across the poems. Moreover, identification of the different types of figurative speeches was done with the purpose of understanding contextual utilization of words to express local beliefs, experiences, norms, values, etc. Finally, results were presented using major theme, and categories supported with quotations. Although, it is difficult to put the translated version of the figurative speeches, the English versions of quotes were used in the result section after translating it from Afan Oromo to English by experts.

## Trustworthiness (rigor)

The trustworthiness of the study findings were ensured through different techniques. First, two investigators analyzed the data. Also, a half day orientation was given to the data collectors. A codebook manual was developed to precisely code the data and enhance credibility. Moreover, the poems were selected from different schools and grade level. This is important to clearly depict the emotions, beliefs, practices and experiences of the local community related

to the malaria prevention and control. In addition, it was used to understand the figurative speeches used in the poems to express a certain phenomenon across the districts.

Second, to ensure transferability, the whole research process, methodology, interpretation of results, and contributions of investigators and data collectors were thickly described. Third, whether study's findings clearly represent the selected poems rather than the belief, theories or biases of the investigators, it was promoted by discussing the findings with experienced researchers working on related topics within the same zonal area. Fourth, researcher self-reflectivity and bracketing is also used to trustworthiness. All the investigators were public health professionals in their educational background and had experience in qualitative research. However, the context of study setting differs from the setting at which the investigators had been worked. Therefore, even if bias is inevitable or unavoidable at any studies, their experience would not lead to a bias that affects the study findings. As much as possible, the subjectivity of the researcher on this study managed by balancing together the data, analytic processes, and findings in such a way that the reader would able to confirm the adequacy of the findings. This background was used to minimize interpretation bias. Fifth, the investigators takes time on the poems to understood more about the emotions, beliefs, practices and experiences of the local community related to the malaria prevention and control, and also figurative speeches used in the poems to express a certain phenomenon. The original copy of the poems (i.e., not translated) were used during the analysis. Sixth, audit trial was also done to ensure conformability and dependability of the study with experienced researchers on qualitative research.

### Ethics approval and consent to participate

This study was approved by an institutional review board, Institute of Health, Jimma University with reference number: IRB 000204/20. Then, support letters were collected from study settings for further approach. The data were stored in a secure cabinet in the department of health, behavior and society. Oral informed consent to access the poems was sought from school directors.

## Results

### Profiles of poetic students and poems

In this study, twenty poems were analyzed to understand the conveyed messages. The ages of students who developed the poems ranged from 12 to17 years old (mean age, 14.3 years). Females and males each contributed ten poems (Table 2). The all poems developed by Afan Oromo language.

**Table 2. Demographic characteristic of the students involved on development of poems to disseminate malaria message in Jimma zone, Oromia, Ethiopia, 2020.**

| Characteristics | Category | Number | Percent (%) |
|---|---|---|---|
| Age category | 10–14 | 16 | 80 |
| | 15–19 | 4 | 20 |
| Sex | Male | 10 | 50 |
| | Female | 10 | 50 |
| Grade level | 5th | 5 | 25 |
| | 6th | 5 | 25 |
| | 7th | 5 | 25 |
| | 8th | 5 | 25 |

### Figurative speeches expressed in the poems to convey belief, norms and practice related to malaria

Different figurative languages were utilized in the poems to convey local beliefs, norms and practices related to malarial diseases. Generally, the study explored four types of figurative speeches, namely simile, metaphors, personification and hyperbole. However, there were over-lapping statements in between these types of figurative speeches. This means that one category of figurative speech might contain other types of figurative speech (s) or there is no demarca-tion in between the different types of figurative speeches. Furthermore, the detail explanations were described below.

### Simile

In the context of this study, simile is denoted as anything related to the malarial perception and practices expressed or compared with other unlikely things instead of direct representa-tions. Introductory words such as like, so and as was used in the poems to denote simile. Across the poems, simile form of figurative speech was mainly used to express distinguishable nature of anopheles mosquito with other insects. It was also used to distinguish the signs and symptoms of malaria with other events.

**Simile of the nature of anopheles mosquito.**   In the poems, the causative agent of malaria, *anopheles mosquito*, was distinguished and expressed as it seems small insect, flies, and vermin. For example, a poem developed, disseminated and communicated by *14 years old*, *male*, *grade 6 from Shebe sembo district expressed*: *"The name of mosquito is anopheles mosquito. . . when you see, it seems like fly."*

**Simile on the malarial disease.**   In the poems it was also expressed that community con-siders malaria as a simple disease that could not lead to severe consequence. It was also empha-sized in the poems that the community was at risk of malarial disease unless immediately sought treatment or apply practical measures. For example, a grade 7 student from Nono-Benja said,

> *". . .malaria is not simple as some people think*
>
> *It kills if you don't go to health facility*
>
> *Let's get up for the fight*
>
> *Don't be lost in the battle"*

**Simile on sign and symptoms of malaria.**   In the poems, the common signs and symp-toms of malaria expressed in similes language were fever (feeling hot), chilling and coldness, and back pain. For example, fever (feeling hot) was expressed like boiling water or hot sun. *"It makes me fevered like that of a strong sunny season. . ." (13 years old, grade 5, male, Limmu-Kosa district,)*

Also, chilling and coldness was expressed in terms of like someone dancing, and shivering during rainy season.

> *"Would you please hold me dawn that I am shivering like a dancer." (15 years old, male, grade 8, Gera district)*
>
> *"Would you please hold me, I will compute dancing." (15 years old, grade 6 male, student, in Gera district)*

*"It makes me a colder and shivers like that of a heavy rainy season." (13 years old, grade 5, male, Limmu-Kosa district,)*

The other sign/symptom of malaria expressed in the simile type of figurative speech was back pain and body weakness. It was expressed like fracture of backbone. A *17 years old, grade 8, from Nono Benja district)*

*"It causes headache also sweeting*

*It causes loss of appetite also vomiting*

*You feel like as your backs become broken. . ."*

## Metaphors

In the context of malaria poems, metaphors were defined as expressions that were used to directly represent perceptions and practices according to mental models of the local community including proverbs, symbols, objects, situations, and phenomena. Therefore, different metaphoric expressions were used related to malaria, anopheles mosquito or the practical measures giving different name and comparing it with closely related objects.

**Metaphoric expression on malaria control and prevention measures.**   In the poems, words/terms such as war and fighting battles were used to represent the readiness and adaptation of task to prevent and eliminate malaria, and its prognosis when reluctantly handled. For example, a grade 7 student from Nono-Benja said,

*". . .malaria is not simple disease as some people think*

*It kills if you don't go to health facility*

*Let's get up for the fight*

*Don't be lost in the battle*

*Oh, people of Jimma rise for the solution*

*Defeat the war against malaria.*

*Why we are beaten while we can win it?"*

Also, the name enemy was also given to the mosquito and malaria. In the poems it was expressed that cleaning environment and breading sites is fundamental act to attack the mosquito. A student of grade 6 from Botor-Tolay said:

*". . .We do not give place to malaria*

*Let's clean up our environment*

*Let's remove mosquito's breeding site*

*So that put our enemy under the control."*

**Metaphoric expression on severity of malaria.**   Different metaphoric expressions were used to denote the severity/threat of malaria on human beings. For example, death related to malaria was described as eaten by malaria, life taken by malaria, and pass of life by malaria. In addition, death-related to malaria was expressed as "being consumed or eaten by malaria." i.e.

to imply that deaths from malaria are premature and preventable, and should be modified through social and behavior changes. An 8th grader female student from Gera expressed,

*"Malaria wants to take the lives of many people*

*Just like yesterday when it consumed the lives of my people..."*

It was also expressed that mosquito is a thief that steals life of human being. It was expressed to denote that unless the peoples protect themselves from malaria by actively using ITN and other preventive measures, the mosquito attack them. In the poems, this condition was expressed that it is easy for a thief to enter someone's house through a door left open. For example, a grade 5 female student, from Shebe-Sombo said that:

*"...Why do we forget caring our life?*

*...Why do not we use the bed nets?*

*So that a thief will not enter the house*

*Through the door that we have opened (read 2X)."*

**Metaphoric expression on ITN.** In the poems, ITN was expressed in different ways. First, it was represented by trap, i.e. to convey a mental picture that rural people use to hunt animals (e.g.: pigs, monkeys, etc that can damage farming). Through this students warned the community and actively use ITN to trap mosquitoes. It was emphasized that as a poorly maintained traps losses to catch those animals, similarly, if peoples improperly use ITN, they lose the game over malaria. A grade 6 male student from Botor-Tolay said,

*"I will tie a trap and spend the night under it,*

*Where do you get me? Why do you try to bite me?*

*If you hang the trap, mosquito don't bite you anymore*

*So, my people don't joke regarding bed net,*

*Utilize it properly, don't pierce and discard it*

*Hanging the trap on our bed, we will capture and trouble it*

*Finally, it will cry anxiously and left in there..."*

Second, the students also expressed that ITN is a treatment for malaria. They emphasized that the community should have to actively use the ITN similar to the drugs prescribed by health professionals to safeguard a family and community from risk of malaria. This is because they mentioned that there was ITN malpractice. It was expressed that people misuse it as a rope and sacks. It also indicated that is foolishness to misuse ITN, and warned the community to consider it as a prescribed treatment. An 8th grader male student from Botor-Tolay said:

*"... Bed net is a treatment that has a chemical that burn malaria..."*

**Metaphoric expressions in the form of proverbs on malaria prevention practices.** In the poems, local beliefs, norms and initiative practices were expressed in the form of proverb. Predominantly, it was used focusing on preventive measures adaptation, especially the active

use of the ITN. For example, 12 years old student from the Shebe—Sembo district was stated the next three ironic expressions. *". . . If someone becomes a disease of himself, what malaria does. . .?"* The students emphasized to denote that the community malpractice or misuse ITN and use it as a rope or to roast maize or other cereals.

The other proverb was: *"If a man breaks the horn of his cow, others cut its neck."* It is a proverb commonly used among Oromo ethnic group to denote that unless everyone is responsible to safely keep his/her property, others do not take care of the property denied or hated by the owner. Therefore, in the context of this study, this metaphoric expression was used to denote that everyone is responsible for his/her health. Through this proverb, students emphasized to disseminate information to the community related to the active use of ITN and reduce risk of malaria.

The other metaphoric expression was *"Through the garden that has no fence or left open by the owner, the thief enters nine times."* This is also a proverb commonly used among Oromo ethnic group to denote that everyone is responsible for his/her own health. In the context of this study, students emphasized to disseminate information that unless the community actively practiced malaria preventive and control practices, they are susceptible and infected with anopheles mosquito, and contract malaria. In this proverb, malaria was also expressed as a thief that enters a house (body). The other metaphoric expression literally stated in the poems by 13 years old student from Limmu-Kosa district was: *"Considering health as simple, fire started to cross the river."* Literally, it means that water can destroy fire however, fire confidently started to cross river. Through this proverb, students emphasized to disseminate information that the community malpractice ITN and contract malaria.

## Personifications

In the context of this study, personification denotes the use of human characters to describe or explain situations or inhuman objects in relation to causes, signs, prevention and treatment of malaria. For example, the students personified anopheles mosquito like a person who plans a day to day activities, carried out activities with great effort to change/improve the one's own life and make people feel sorrow, cry or ban from the earth. It was also personified like a person who can build his/her residence home/house.

> *"This mosquito has many works, what she does, she does her works and by sitting on a person she changes her life (daily life)."* (A 15 years old, grade 6 male, student, school of L/Botor, in Botor-Tolay district)

> *"It (malaria) finished the community, which don't fear the creator (God) and also in order to bans and sorrow/cry the people it (the* anopheles mosquito*) works very hard."* (15 years old, grade 6 male, in Gera district)

> *"We were damaged/injured by malarial diseases and he (the* anopheles mosquito*) built his house and reproduce in our body."* (16 years old, grade 8, male, student in Botor–Tolay district)

## Hyperbole

In the context of this study, hyperbole is a statement made emphatic by overstatement of the malarial perception and practices conveyed in the poem message. Thus, different hyperbolic statements were conveyed from the poem messages developed by the students.

**Hyperbole on mosquito and its biting behavior.**   The mosquito was expressed that is has a very wide and sharp mouth, and enters deep root of a body (i.e., bloodstream) while biting. For example, one 8th grader female student from Gera district, said:

*". . .she (mosquito) has a mouth that is very wide*

*Through which she enters a person's deep root during biting. . ."*

*"It has a sharp mouth to feed on human blood.*

*It moves from person to person looking a normal man.*

*It hurts a lot though it looks harmless when it comes to you."*

**Hyperbole on the severity or seriousness of malarial disease.**   It was expressed that malarial disease is a severe disease than HIV. This is because it was mentioned that HIV would give some period of time for a person to live in life. However, malaria leads to a sudden death and kills the whole human being from the earth. For example, 16 *years old*, *grade 8*, *female*, *from Limmu-Kosa district expressed*:

*". . .It (malarial disease) is more severe than from HIV*

*HIV can give days -years for the people*

*But malaria doesn't give us time."*

*It causes sudden deaths and leaves the earth without human being."*

**Hyperbole on malaria prevention practices.**   A hyperbolic expression was also used to express the effectiveness of ITN and IRS. It was expressed that utilizing ITN and IRS would diminish the blood stream of the malaria. This is to mean that active use of ITN and IRS kills mosquito and reduce risk of malaria. Thus, the students emphasized the benefit of using these preventive measures.

*"By utilizing ITN and IRS, we will diminish its (malaria) blood stream." ((15 years old, grade 6, male, in Gera district)*

## Discussion

Obviously, there are a lot of figurative speeches used in poems to convey messages in any artistic communication [11, 18, 19]. Predominantly, this study explored four figurative speeches, namely simile, metaphor, personification and hyperbole, used in the poems to convey messages related to malarial perceptions, beliefs, norms and practices to prevent malaria.

Simile form of figurative speeches expression was used in the poems developed, disseminated and communicated by primary school students to indicate the perception, beliefs, norms and practices related to malaria. The community considered the anopheles mosquito as a small insect, flies, and vermin despite it causes malaria and results in death. This underscores there is a need to design a health education program and conduct social and behavioral change communication interventions to change the risk perception of community towards anopheles mosquito and malarial disease, and active use of ITN and practice of other preventive measures like household spraying of IRS, remove stagnant women, clean the surrounding environment.

The sign and symptoms of malaria were also expressed in simile form of figurative speeches. The students expressed contents related to the signs and symptoms of malaria distinguishing with the locally understandable conditions and or phenomenon in their local surroundings. Therefore, this form of expression, where the students used a distinguishing phenomenon

from their surrounding environment or experience to convey messages is very fundamental and use of local contexts and mental heuristics are effective ways of communicating perceptions, attitudes, and promoting practices [18, 20–23].

Metaphoric expression was the other commonly used figurative speeches expressed in the poems. The study found that malaria and anopheles mosquito is an enemy of the community and there is a need to apply all the preventive measures or practices towards malaria elimination and control (i.e., war at a fighting battle). However, it was found that there was misuse or malpractice of ITN among the community members that underscores the need to design health education program and conduct social and behavioral change communication interventions to bring the desired behavioral change.

On the other hand, the severity of malaria was also metaphorically expressed by giving names like thief that steals life, consumes peoples and destroys all human beings from the world unless preventive and control measures. This implies that controlling, preventing and eliminating malaria is the responsibility of every individual. However, the study found that the community members did not carried out all preventive measures, especially misuse or malpractice ITN. Therefore, this underscores that there is a need to conduct social and behavioral change communication interventions to bring the intended behavior towards use of ITN. Therefore, the students developed to convey a message for the community to use the ITN properly and keep from malaria, and they are responsible to keep their health. Therefore, students were used metaphoric expression giving different naming to malaria, anopheles mosquito or the practical measures, and comparing with closely related symbols, objects, situations, and phenomena according to mental models of the local community [18, 20–23].

The study also found that ITN was metaphorically expressed as a trap in the local language that the community can early conceptualize its benefit. It was found that properly handled ITN is used to catch the anopheles mosquito and reduce risk of infection. This implies that being infected by the mosquito and develop a malarial disease depends on the proper use and the quality of the ITN. ITN was also named as a prescribed drug to emphasize that active use of the ITN is fundamental to safeguard a family and community from risk of malaria. Therefore, students were used metaphoric expression giving different naming to malaria, anopheles mosquito or the practical measures, and comparing with closely related symbols, objects, situations, and phenomena according to mental models of the local community [18, 20–23]. However, it was also found that the community did not use it properly. This underscores the need to conduct social and behavioral change communication interventions to change the behavior of community and properly use the ITN.

Personification is the other form of figurative speech expressed in the poems to convey malaria messages. The study found that the anopheles mosquito and malaria was given human characters who can do evil things while looking good in wiles (e.g: some people attack others while laughing at them) in their act of biting, feeding, and transmitting malaria. This implies that the community should have to protect themselves from malaria through active use of ITN and other measures. Therefore, students were used metaphoric expression giving different naming to malaria, anopheles mosquito or the practical measures, and comparing with closely related symbols, objects, situations, and phenomena according to mental models of the local community [18, 20–23].

The study also found a hyperbolic expression to overstatement of the malarial perception and practices conveyed in the poem message. The anopheles mosquito and its biting behavior was expressed as the one which has a very bad, wide and sharp mouth through which it bits and enters in the deep root of a person's body (i.e., to mean bloodstream). Also, malaria was expressed in terms of diseases that is serious than HIV. This was used to imply that malaria is not a simple disease rather it is a severe or serious diseases. This is to imply that malaria is a very

severe disease and unless treated immediately, it leads to a sudden death. Therefore, the students used overstatement to emphasize in conveying a message and to disseminate the perceived severity/seriousness of malaria to the community, and alarm them to control and prevent malaria, and also to seek health care immediately after sign/symptom recognition [18, 20–23].

## Strength and limitation of the study

To the best of the investigator's knowledge, this is the first research conducted on this issue in Ethiopia. The contents reported in this work may emphasize on connotative as well as hidden meanings. Therefore, the findings will be used as an input for any communication intervention aimed at the community level. However, due to limited literatures, the findings were not well discussed. On the other hand, the poems were selected from schools under the school-based malaria project aimed to advance community knowledge and practices. Thus, the contents of any poems generated outs of the project site were not produced. Moreover, the poems were developed by primary school students; from nonprofessional or inexperienced source. However, there was very essential and fundamental concepts expressed in terms of figurative speeches that could motivate, promote and put into practice malarial prevention and control measures by the community.

## Conclusions

The study found figurative speeches such as similes (i.e., to express the anopheles mosquito, and sign and symptoms of malaria), metaphors (i.e., to express malaria, severity/seriousness of malaria and ITN), personification (i.e., to express anopheles mosquito and malaria), and hyperbole (i.e., to express anopheles mosquito, severity of malaria and exaggerated effect of ITN and IRS, and to express the relationship between persons ITN malpractice and its effect on their health in the form of proverbs). The students used the figurative speeches in their poems to convey messages related to malarial perceptions, beliefs, norms and practices of the local community to prevent malaria. This implies that program planners, implementer's or different stakeholders interested to implement a community-based intervention should have to conceptualize the local norms, beliefs, values, perception and practices, and use different figurative speeches to convey messages and convince the local community to bring the desired or intended behavioral change.

## Acknowledgments

We would like to thank and express our heartfelt gratitude to all individuals who participated in the study: data collectors, school administrative officials, and language experts who supported in dealing with poetic literatures.

## Author Contributions

**Conceptualization:** Zewdie Birhanu, Yohannes Kebede.

**Data curation:** Kasahun Girma Tareke, Abdu Hayder, Yohannes Kebede.

**Formal analysis:** Kasahun Girma Tareke, Abdu Hayder, Firanbon Teshome, Yohannes Kebede.

**Funding acquisition:** Kasahun Girma Tareke, Abdu Hayder, Zewdie Birhanu, Yohannes Kebede.

**Investigation:** Kasahun Girma Tareke, Zewdie Birhanu, Yohannes Kebede.

**Methodology:** Kasahun Girma Tareke, Abdu Hayder, Firanbon Teshome, Yohannes Kebede.

**Project administration:** Kasahun Girma Tareke, Zewdie Birhanu, Yohannes Kebede.

**Resources:** Kasahun Girma Tareke, Yohannes Kebede.

**Software:** Kasahun Girma Tareke, Abdu Hayder, Firanbon Teshome, Yohannes Kebede.

**Supervision:** Kasahun Girma Tareke, Yohannes Kebede.

**Validation:** Kasahun Girma Tareke, Abdu Hayder, Firanbon Teshome, Yohannes Kebede.

**Visualization:** Kasahun Girma Tareke, Yohannes Kebede.

**Writing – original draft:** Kasahun Girma Tareke.

**Writing – review & editing:** Kasahun Girma Tareke, Abdu Hayder, Firanbon Teshome, Zewdie Birhanu, Yohannes Kebede.

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
