## [Decision Letter · Decision Letter 0]

22 Jun 2021

PONE-D-21-03762

Students’ figurative communication of malaria messages, belief, norms, and practices in Oromia, Ethiopia: A qualitative content analysis approach

PLOS ONE

Dear Dr. Tareke,

Thank you for submitting your manuscript to PLOS ONE. After careful consideration, we feel that it has merit but does not fully meet PLOS ONE’s publication criteria as it currently stands. Therefore, we invite you to submit a revised version of the manuscript that addresses the points raised during the review process.

The reviewers raised a number of concerns with the manuscript, including issues with grammar, structure and presentation. They also had concerns about the methodological approach and discussion of the results. Their comments can be viewed in full, below.

We look forward to receiving your revised manuscript.

Kind regards,

Natasha McDonald, PhD

Associate Editor

PLOS ONE

Journal Requirements:

"Institute of Health, Jimma University"

Reviewers' comments:

Reviewer's Responses to Questions

**Comments to the Author**

1. Is the manuscript technically sound, and do the data support the conclusions?

Reviewer #1: Partly

Reviewer #2: Partly

2. Has the statistical analysis been performed appropriately and rigorously? 

Reviewer #1: N/A

Reviewer #2: N/A

3. Have the authors made all data underlying the findings in their manuscript fully available?

Reviewer #1: No

Reviewer #2: No

4. Is the manuscript presented in an intelligible fashion and written in standard English?

Reviewer #1: No

Reviewer #2: No

5. Review Comments to the Author

Reviewer #1: The manuscript reports a qualitative study that examined the content of poems created by students of primary schools in Oromia, Ethiopia. The study analyzed how students used figurative languages in their poems to express the vector mosquitoes, signs, symptoms, risk, of malaria and its preventive measures. There have been few similar studies in the area of malaria control. Therefore, the findings of the study can contribute to better designing interventions for malaria control. However, the manuscript needs to be revised, as it failed to address major points. The authors should consider the following points to improve the manuscript.

1. Information on the school-based SBCC intervention in the study should be provided: How and why did the students create poems? How were the students educated about malaria?

2. The generalizability of the findings should be discussed. This is because the study samples were collected only from the intervention schools.

3. Are there any similar studies to which the authors referred to develop the data analysis strategies and to validate the results? What literature was used for planning the methods?

4. Writing opinions in the results section should be avoided. For example, in the second paragraph of the “Figurative speeches….malaria”, the authors mentioned “Therefore, anyone who can go through this work should have to understand…”.

5. It would be better to divide the second paragraph of the discussion section (Simile is…) into two paragraphs, as the paragraph includes two different topics such as anopheles and sign/symptom.

6. Importantly, potential risk/negative consequence of “figurative communication” should be discussed. For example, “anopheles mosquito was expressed like…flies” and “malaria was expressed in terms of diseases that is serious than HIV”.

7. In the limitation section, the authors mentioned that “due to limited literature, the findings were not well discussed”. I do not fully agree with the authors, because although there are few studies that examined figurative languages for malaria control, there are a number of studies that examined figurative languages for other topics.

8. In the conclusion section, the authors stated that “This implies that use of different figurative speeches is very important….”. What does support the statement? Why did the authors consider the figurative speeches very important? Was the conclusion based on the results?

Reviewer #2: This an interesting and important paper in the area of of social and behavioral change communication. These findings could form part of the information or knowledge that contributes towards literature on how messages for SBCC could be developed.

The paper can be improved considerably. following are some of the elements that would need to be addressed in order to improve it:

Pay attention to grammar. Important things, beginning from the title itself, the word “Belief” should be “beliefs”.

Important to revisit the paper to improve the grammar which would enhance its readability. An important way to achieve this is replacing the long and winding sentences such as the second paragraph in the introduction section with shorter ones. Similarly, the second sentence could be improved with appropriate punctuation. Consider these for the entire paper.

The first sentence in the third paragraph should be revisited as well. A poem is not a form of malaria message.

Revisit the second line of the fourth paragraph specifically the sentence “…, and the primary students were developed and disseminated malaria messages…” it does not make sense.

In the data analysis section, there is content talking about selection of the 20 poems. This information must be removed. The sample and sampling section should have covered this content / information.

As already mentioned, the entire paper should be revisited to improve grammar. In the analysis section, sentences like: “Then, the coding of the whole poems...” should be corrected. Similarly, statement like: “The coding system repeated the four times after the draft code book was developed”. These and various other statements should be revisited.

The data analysis should focus on the specific area of interest (figurative language) that was analyzed than to provide details such as typical analysis done to poems like the structure of the poem and setting of the poem etc. if this did not form part of the eventual themes that are central to this paper.

Adopt a consistent writing / format style. For example, SIMILE and HYPERBOLE are in capitals while Metaphors, in small letters. In this case it seems the main themes were supposed to be in CAPITALS while the sub-themes in small letters. If that is the case, then be consistent.

The discussion could be discuss more the findings of the study mainly. For example, the opening statements talks about many figurative speeches that the study did not find or work with. While the statement is meant to provide a general picture of figurative speeches that exist, it does not provide a general statement that properly represent the findings of this study.

The discussion should NOT be a re-presentation of the results. Rather it should interpret and describe the significance of these findings in light of what was already known about the issue / problem being investigated and to explain any new understanding or insights that emerged as a result of this study. This is not coming out clearly. The discussion is largely explaining how the different figurative speeches were used. Ideally it should probably talk about the relevance of using personification in the fight against malaria and linking that with studies that could show the importance of such kind of information. The last paragraph that discuss the hyperbolic expression is a good example of how best to discuss these findings.

The study limitation acknowledges the paucity of literature in Ethiopia on this topic and attributes the “inadequate” discussion to this. However, the discussion could use literature from elsewhere if available. Furthermore, even without literature, the discussion could be improved considerably.

The conclusion should be revisited. While some elements of what a conclusion is exist, it can be improved. The conclusion should NOT be a summary of the findings, rather it should wrap up the author’s ideas around the topic and leave the reader with a strong final impression of what the study implies.

6. PLOS authors have the option to publish the peer review history of their article (what does this mean?). If published, this will include your full peer review and any attached files.

Reviewer #1: **Yes: **Daisuke Nonaka

Reviewer #2: **Yes: **Eric Umar

---

## [Author Response · Author response to Decision Letter 0]

6 Aug 2021

Response to reviewers

Dear PLOS ONE academic editor and reviewer,

We want to express our deepest gratitude for reviewing and providing your constructive comments to us on a manuscript entitled “Students’ figurative communication of malaria messages, belief, norms, and practices in Oromia, Ethiopia: A qualitative content analysis approach” submitted to PLOS ONE journal. We are very much grateful for the editor and reviewers’ time and willingness to review the manuscript. The authors found the comments of the editor and reviewers very important. The authors were really very happy to address the comments raised by the editor and reviewer for this manuscript. We have seriously considered the comments and have gone through the entire body of the manuscript to make necessary editorial corrections and clarifications to the concerns. Corrections are shown by highlighting the texts in yellow color in the main document. Please find below our responses based on each comment. 

Response to reviewer 1: 

Thank you very much for your comments. We have amended as per your constructive comments and please find below. 

1. Information on the school-based SBCC intervention in the study should be provided: How and why did the students create poems? How were the students educated about malaria?

Response: Thank you so much for the comment. The students were used as key actors for the social and behavioral change communication aiming to develop and effectively and efficiently disseminate malaria messages to the wider community to improve the capacity, knowledge, and decision-making skills that help to promote health and prevent diseases. The intervention was conducted in the study setting from 2017 to 2019. At the beginning of the intervention, training was given to the students and focal teachers. Then, under the supervision of the focal teachers, students developed malarial messages in the form of poems and disseminated it to the community through different means. Introduction, page 3, line 66-85.

2. The generalizability of the findings should be discussed. This is because the study samples were collected only from the intervention schools.

Response: Thank you so much. It is obvious that in qualitative research, the study findings are not generalized to the other settings rather transferred to similar contexts. Therefore, the issue of transferability was discussed under trustworthiness section. Method, page 6 &7, line 178-172. In addition, we have described that the poems were selected from schools under the school-based malaria project aimed to advance community knowledge and practices. Thus, the contents of any poems generated out of the project site were not produced. Strength and Limitation of the Study, page 16, line 473-475

3. Are there any similar studies to which the authors referred to develop the data analysis strategies and to validate the results? What literature was used for planning the methods?

Response: Really, thank you for this comment. We have not yet got studies conducted on similar topics with similar population. However, we used a poetic analysis approach described in different literatures. For example, we have used reference number 18-25 to plan and guide the analysis and method. Method, page 6, line 150

4. Writing opinions in the results section should be avoided. For example, in the second paragraph of the “Figurative speeches….malaria”, the authors mentioned “Therefore, anyone who can go through this work should have to understand…”

Response: We have amended it as per your comment. Result, page 8, line 213-219.

5. It would be better to divide the second paragraph of the discussion section (Simile is…) into two paragraphs, as the paragraph includes two different topics such as anopheles and sign/symptom.

Response: We have amended it as per your comment. Discussion, page 14, line 408.

6. Importantly, potential risk/negative consequence of “figurative communication” should be discussed. For example, “anopheles mosquito was expressed like…flies” and “malaria was expressed in terms of diseases that is serious than HIV”.

Response: Thank you so much for this comment. Yes, it needs some explanation and we have provided the description that it causes misconception and confusion among the audience. Discussion, page 16, line 466, 469.

7. In the limitation section, the authors mentioned that “due to limited literature, the findings were not well discussed”. I do not fully agree with the authors, because although there are few studies that examined figurative languages for malaria control, there are a number of studies that examined figurative languages for other topics.

Response: Thank you so much. You are right a lot of papers are already available on different topics and it is not mandatory to get similar article. Therefore, we have amended it and used different articles to discuss the findings. Strength and limitation of the study, page 16, line 471-479

8. In the conclusion section, the authors stated that “This implies that use of different figurative speeches is very important….”. What does support the statement? Why did the authors consider the figurative speeches very important? Was the conclusion based on the results?

Response: It is true that we made a mistake while concluding the study findings. We have amended it to be consistent with the findings. We concluded that the study found that the students conveyed messages related to malarial perceptions, beliefs, norms, and practices of the local community to prevent and control malaria through different types of figurative speeches; simile, metaphor, personification, and hyperbole. Predominantly, the students expressed in their poems that there was malpractice of malaria prevention and control measures, especially on the use of ITN among the community. This underscores that there is a need to design health education programs and conduct SBCC interventions to properly practice malaria prevention and control measures. Conclusions, page 16, line 481-487

Response to reviewer 2: 

Response: Thank you so much for your constructive comments. We accepted all of your comments are valuable enough to improve the quality of the manuscript. Majorly, you have raised the grammatical and typographical errors. Thus, accepted your comments and amended it throughout the manuscript. We have also changed the word “Belief” in the title part to “beliefs”. Title, page 1, line 1 

2. Important to revisit the paper to improve the grammar which would enhance its readability. An important way to achieve this is replacing the long and winding sentences such as the second paragraph in the introduction section with shorter ones. Similarly, the second sentence could be improved with appropriate punctuation. Consider these for the entire paper.

Response: We have amended it as per your constructive comments throughout the manuscript. . 3. The first sentence in the third paragraph should be revisited as well. A poem is not a form of malaria message.

Response: Yes, it it true that the poem is not a form of malaria message. We have rephrased it accordingly. Introduction, page 3, line 82-85 

4. Revisit the second line of the fourth paragraph specifically the sentence “…, and the primary students were developed and disseminated malaria messages…” it does not make sense.

Response: We have rephrased it accordingly. Introduction, page 3, paragraph 3, line 79-82

5. In the data analysis section, there is content talking about selection of the 20 poems. This information must be removed. The sample and sampling section should have covered this content / information.

Response: It is true that this information should be included under sample and sampling technique. Thus, we have removed it from the analysis part. Method, data analysis, page, 6

6. As already mentioned, the entire paper should be revisited to improve grammar. In the analysis section, sentences like: “Then, the coding of the whole poems...” should be corrected. Similarly, statement like: “The coding system repeated the four times after the draft code book was developed”. These and various other statements should be revisited.

The data analysis should focus on the specific area of interest (figurative language) that was analyzed than to provide details such as typical analysis done to poems like the structure of the poem and setting of the poem etc. if this did not form part of the eventual themes that are central to this paper.

Response: We have accepted your comments and amended all of it. Method, data analysis, page 6

7. Adopt a consistent writing / format style. For example, SIMILE and HYPERBOLE are in capitals while Metaphors, in small letters. In this case it seems the main themes were supposed to be in CAPITALS while the sub-themes in small letters. If that is the case, then be consistent.

Response: Thank you for your comments again. It is true that we made a mistake while writing it. We have amended it typing as Simile, Metaphor, Personification and Hperbole. Result, page 8-13, line 221, 260, 345 and 365

8. The discussion could be discuss more the findings of the study mainly. For example, the opening statements talks about many figurative speeches that the study did not find or work with. While the statement is meant to provide a general picture of figurative speeches that exist, it does not provide a general statement that properly represent the findings of this study.

Response: Yes, it is true. Our assumption was to make an introductory sentence rather than indicating all types of figurative speech. Therefore, we have corrected made it consist with the current study findings. Discussion, page 14, line 397 and 398

9. The discussion should NOT be a re-presentation of the results. Rather it should interpret and describe the significance of these findings in light of what was already known about the issue / problem being investigated and to explain any new understanding or insights that emerged as a result of this study. This is not coming out clearly. The discussion is largely explaining how the different figurative speeches were used. Ideally it should probably talk about the relevance of using personification in the fight against malaria and linking that with studies that could show the importance of such kind of information. The last paragraph that discuss the hyperbolic expression is a good example of how best to discuss these findings.

Response: Thank you again for your comments. We have amended almost all parts of the discussion following your comments. Discussion, page 14-16. 

10. The study limitation acknowledges the paucity of literature in Ethiopia on this topic and attributes the “inadequate” discussion to this. However, the discussion could use literature from elsewhere if available. Furthermore, even without literature, the discussion could be improved considerably.

Response: Thank you so much. You are right a lot of papers are already available on different topics and it is not mandatory to get similar article. Therefore, we have amended it and used different articles to discuss the findings.

11. The conclusion should be revisited. While some elements of what a conclusion is exist, it can be improved. The conclusion should NOT be a summary of the findings, rather it should wrap up the author’s ideas around the topic and leave the reader with a strong final impression of what the study implies.

Response: It is true that our conclusion did not fully represent the study findings. However, following your constructive comments, we have revised it and we concluded that the study found that the students conveyed messages related to malarial perceptions, beliefs, norms, and practices of the local community to prevent and control malaria through different types of figurative speeches; simile, metaphor, personification, and hyperbole. Predominantly, the students expressed in their poems that there was malpractice of malaria prevention and control measures, especially on the use of ITN among the community. This underscores that there is a need to design health education programs and conduct SBCC interventions to properly practice malaria prevention and control measures. Conclusions, page 16, line 481-487

Saying this, I hope that the comments provided by the reviewer were addressed and the manuscript would meet the high standards of your journal. Therefore, am looking forward to receive a favorable response from you regarding the acceptance of the manuscript.

Sincerely yours

Kasahun Girma Tareke (corresponding author)

Address: Department of Health, Behavior and Society, Faculty of Public Health, Institute of Health, Jimma University, Jimma, Ethiopia

E-mail: kasahungirmadera@gmail.com; girma.tareke@ju,edu.et; 

Phone: +251 919375374

---

## [Decision Letter · Decision Letter 1]

27 Apr 2022

PONE-D-21-03762R1Students’ figurative communication of malaria messages, beliefs, norms, and practice s in Oromia, Ethiopia: A qualitative content analysis approachPLOS ONE

Dear Dr. Tareke,

Thank you for submitting your manuscript to PLOS ONE. After careful consideration, we feel that it has merit but does not fully meet PLOS ONE’s publication criteria as it currently stands. Therefore, we invite you to submit a revised version of the manuscript that addresses the points raised during the review process.

 Kindly address the reviewer comments before a decision can be made. Please submit your revised manuscript by Jun 11 2022 11:59PM. If you will need more time than this to complete your revisions, please reply to this message or contact the journal office at plosone@plos.org. Please include the following items when submitting your revised manuscript:A rebuttal letter that responds to each point raised by the academic editor and reviewer(s). You should upload this letter as a separate file labeled 'Response to Reviewers'.A marked-up copy of your manuscript that highlights changes made to the original version. You should upload this as a separate file labeled 'Revised Manuscript with Track Changes'.An unmarked version of your revised paper without tracked changes. You should upload this as a separate file labeled 'Manuscript'.If applicable, we recommend that you deposit your laboratory protocols in protocols.io to enhance the reproducibility of your results. Protocols.io assigns your protocol its own identifier (DOI) so that it can be cited independently in the future. For instructions see: https://journals.plos.org/plosone/s/submission-guidelines#loc-laboratory-protocols. Additionally, PLOS ONE offers an option for publishing peer-reviewed Lab Protocol articles, which describe protocols hosted on protocols.io. Read more information on sharing protocols at https://plos.org/protocols?utm_medium=editorial-email&utm_source=authorletters&utm_campaign=protocols.

We look forward to receiving your revised manuscript.

Kind regards,

Kingston Rajiah

Academic Editor

PLOS ONE

Journal Requirements:

Reviewers' comments:

Reviewer's Responses to Questions

**Comments to the Author**

1. If the authors have adequately addressed your comments raised in a previous round of review and you feel that this manuscript is now acceptable for publication, you may indicate that here to bypass the “Comments to the Author” section, enter your conflict of interest statement in the “Confidential to Editor” section, and submit your "Accept" recommendation.

Reviewer #1: All comments have been addressed

Reviewer #3: (No Response)

2. Is the manuscript technically sound, and do the data support the conclusions?

Reviewer #1: Yes

Reviewer #3: Partly

3. Has the statistical analysis been performed appropriately and rigorously? 

Reviewer #1: N/A

Reviewer #3: No

4. Have the authors made all data underlying the findings in their manuscript fully available?

Reviewer #1: No

Reviewer #3: Yes

5. Is the manuscript presented in an intelligible fashion and written in standard English?

Reviewer #1: Yes

Reviewer #3: Yes

6. Review Comments to the Author

Reviewer #1: (No Response)

Reviewer #3: Title: Students’ figurative communication of malaria messages, belief, norms, and practices in Oromia, Ethiopia: A qualitative content analysis approach.

Comment to the authors

Dear authors’ thank you for submitting this nice manuscript.

Here under some comments about your manuscript.

Why not quantitative and qualitative study? I think that quantitative study includes large sample size and helps to give detail interpretations about the case? Here you have used purposive sampling techniques, due to this your sample size is very small. If you use random sampling technique, it helps you to study every hidden information about the case.

You have used purposive sampling, but in your manuscript you are talking about the strata. Why? “This is because they are categorized under the high-medium-malaria-burden strata.” Justify it.

If you are using the purposive sampling technique, what is your reason to select those schools, try to justify it clearly?

In table 1, you have stated the sample allocation, do you think that is it proportional? I am sure that your answer is no, because there is no equal number of poems in each school.

From which class students do you taken the sample, from grade 5, 6, 7 or 8? Because students in each class has different understandings about the case? Justify it.

What does it mean that “Note: Equal of 5 poems will be considered from grade 5, 6, 7 and 8” in table 1. Explain it clearly.

Do you believe that this sample is the representative of the Jimma Zone primary school students, or can you conclude about the cases in Jimma zone based on this sample?

In you result part you have some descriptive statistics “The ages of students who developed the poems ranged from 12 to17 years old (mean age, 14.3 years). Females and males each contributed ten poems.” But, it is not presented in the tabular form, try to present it in tabular form. Not only this, try to put all results that you have collected like age of students and sex of students.

In general, you missed study designing and sampling techniques in your manuscript revise it seriously.

7. PLOS authors have the option to publish the peer review history of their article (what does this mean?). If published, this will include your full peer review and any attached files.

Reviewer #1: **Yes: **Daisuke Nonaka

Reviewer #3: No

---

## [Author Response · Author response to Decision Letter 1]

3 May 2022

Rebuttal Letter 

Dear editor and reviewers, 

Thank you for your constructive comments you provided for us on the manuscript entitled with “Students’ figurative communication of malaria messages, belief, norms, and practices in Oromia, Ethiopia: A qualitative content analysis approach”. We are very much grateful for the editor and reviewers’ time and willingness to review the manuscript. The authors found the comments of the editor and reviewers very important. The authors were really very happy to address the comments raised by the editor and reviewers for this manuscript. We have seriously considered the comments and have gone through the entire body of the manuscript to make necessary editorial corrections and clarifications to the concerns. Corrections are shown by highlighting the texts in yellow color in the main document.

Response to the Reviewer 3 comments:

Comment 1: Why not quantitative and qualitative study? I think that quantitative study includes large sample size and helps to give detail interpretations about the case? Here you have used purposive sampling techniques, due to this your sample size is very small. If you use random sampling technique, it helps you to study every hidden information about the case.

Response: Thank you very much the reviewer. Yes, it was possible to conduct a quantitative study given that it involves large sample size and make the study findings generalized. However, there are issues that demand qualitative study to gain indepth insights about a certain phenomenon. For example, the current study among those health issues that needs contextual interpretation of the figurative speeches or metaphors. Therefor, the nature of our research questions made us to study the pehenomenon using qualitative content analysis approach.

Comment 2: You have used purposive sampling, but in your manuscript you are talking about the strata. Why? “This is because they are categorized under the high-medium-malaria-burden strata.” Justify it.

Response: Thank you agein for your comment. As it was mentioned in the manuscript, a social and behavioral change communication intervention packages were conducted as selected districs of Jimma zone, oromia, Ethiopia from 2017-2019. The reason why the intervention was conducted at those districts were that malaria endemicity was high and medium, and the intervention aimed to change the behavior of the community through school community engagement.

Comment 3: If you are using the purposive sampling technique, what is your reason to select those schools, try to justify it clearly?

Response: Thank you again for your noce comments. We accepted as this comment was valuable but forgotten in our manuscript. The schools were selected based on criteria’s such as having high number of students enrolled in the school (ranged from 440-1450), being located at high malaria endemic areas, feasibility in terms of distance and active engagement and better involvement of students in producing poems. Method section, page 5, line 132-135

Comment 4: In table 1, you have stated the sample allocation, do you think that is it proportional? I am sure that your answer is no, because there is no equal number of poems in each school.

Response: Infact, the principle of purposive sampling is doesnot mean or consistent with proportional allocation. This is because is because participant recruitment based on purposive sampling technique depends on the richness of information and criterialss. For example, in the method section, page 5, line 135-143 was mentioned that “A purposive sampling technique was used to select poems from school-based malaria SBCC documentation based on like richness, relevancy and data diversity pertinent to the research question. Specific criteria’s used to select the poems were presence of at least three behavioral constructs (from knowledge, attitude, risk- perception, self-efficacy, response -efficacy, and practice ) through roughly reading; presence and readability of the poems, length of the poems (i.e., at least two pages) and presence at least one type of figurative speech. Accordingly, 20 poems were selected from 20 selected schools across the districts. A sample of poems was selected from different schools across the districts and students grades to maximize data triangulation.” This means that the samples were not proportionally allocated rather determined, by default, by criterials mentioned above. 

Comment 5: From which class students do you taken the sample, from grade 5, 6, 7 or 8? Because students in each class has different understandings about the case? Justify it.

Response: We took sample of poems from all 5-8 classes. Yes. It is true that there might be different level of understanding about the health issue across the grade levels. However, it the introduction section, page 2, line 72-80 of the manuscript, it was mentioned that adequate training was given to all students from 5-8 grade levels to made their understanding close enough as much as possible. 

Comment 6: What does it mean that “Note: Equal of 5 poems will be considered from grade 5, 6, 7 and 8” in table 1. Explain it clearly.

Response: Thank you for the comment. It was to mean five poems were purposively selected from grade 5, 6, 7 and 8. Method section, page 5, table 1 

Comment 7: Do you believe that this sample is the representative of the Jimma Zone primary school students, or can you conclude about the cases in Jimma zone based on this sample?

Response: Thank you gain for the comment. Infact, in the paradigm of quantitative research the sample size was very small to represent the sstudy population and mage the study finding generalize. However, in case of qualitative research, this is different. The nature of qualitative research is to recruit small sample size from different contects (setting, population, time, etc). Therefore, in this study, poems prepared by students from grade 5, 6, 7 and 8 were analyzed. In addition, the poems were selected from different schools or districs. Therefore, only sample size doesnot deteremine representativeness in qualitative research. Rather thickly describing the research process and contexts ensure dependability and transferability of the findings. These issues were described in the manuscript, under trustworthiness section, page 6 and 7, line 176-201. 

Comment 8: In you result part you have some descriptive statistics “The ages of students who developed the poems ranged from 12 to17 years old (mean age, 14.3 years). Females and males each contributed ten poems.” But, it is not presented in the tabular form, try to present it in tabular form. Not only this, try to put all results that you have collected like age of students and sex of students.

Response: Thank you so much! We accepted as it was and amended it accordingly. Result section, page 8, Table 2

Comment 9: In general, you missed study designing and sampling techniques in your manuscript revise it seriously.

Response: It is obvious that qualitative study has different approaches. Different schoolars donot use the term ‘study design’ is qualitative study rather’study approaches’. This is because it fouses on an indepth or insight interpretation of meanings. Therefore, we used a qualitative content analysis approach to interprete figururative speeches used in the poems. Method section, page 4, line 122-127. 

Saying this, we hope that the comments provided by the reviewer were addressed and the manuscript would meet the high standards of your journal. Therefore, we are looking forward to receive a favorable response from you regarding the acceptance of the manuscript.

Sincerely yours

Kasahun Girma Tareke (corresponding author)

Address: Department of Health, Behavior and Society, Faculty of Public Health, Institute of Health, Jimma University, Jimma, Ethiopia

E-mail: kasahungirmadera@gmail.com; girma.tareke@ju,edu.et; 

Phone: +251 919375374

Thank you!!!

---

## [Editor Report · Decision Letter 2]

10 May 2022

Students’ figurative communication of malaria messages, belief, norms, and practices in Oromia, Ethiopia: A qualitative content analysis approach

PONE-D-21-03762R2

Dear Dr. Tareke,

We’re pleased to inform you that your manuscript has been judged scientifically suitable for publication and will be formally accepted for publication once it meets all outstanding technical requirements.

Kind regards,

Kingston Rajiah

Academic Editor

PLOS ONE
---

## [Editor Report · Acceptance letter]

12 May 2022

PONE-D-21-03762R2 

Students’ figurative communication of malaria messages, belief, norms, and practices in Oromia, Ethiopia: A qualitative content analysis approach 

Dear Dr. Tareke:

I'm pleased to inform you that your manuscript has been deemed suitable for publication in PLOS ONE. Congratulations! Your manuscript is now with our production department. 

Kind regards, 

on behalf of

Associate Professor Kingston Rajiah 

Academic Editor

PLOS ONE